# Blood Stream Microbiota Dysbiosis Establishing New Research Standards in Cardio-Metabolic Diseases, A Meta-Analysis Study

**DOI:** 10.3390/microorganisms11030777

**Published:** 2023-03-17

**Authors:** Mohsan Ullah Goraya, Rui Li, Liming Gu, Huixiong Deng, Gefei Wang

**Affiliations:** Guangdong Provincial Key Laboratory of Infectious Diseases and Molecular Immunopathology, Shantou University Medical College, Shantou 515041, China

**Keywords:** circulating bacteria, blood microbiota, dysbiosis, 16S rDNA, 16S rRNA, cardiovascular diseases, metabolic diseases, diabetes

## Abstract

Aims: Scientists have recently discovered a link between the circulating microbiome and homeostasis, as well as the pathogenesis of a number of metabolic diseases. It has been demonstrated that low-grade chronic inflammation is one of the primary mechanisms that has long been implicated in the risk of cardio-metabolic disease (CMDs) and its progression. Currently, the dysbiosis of circulating bacteria is considered as a key regulator for chronic inflammation in CMDs, which is why we have conducted this systemic review focused on circulating bacterial dysbiosis. Methods: A systemic review of clinical and research-based studies was conducted via PubMed, Scopus, Medline, and Web of Science. Literature was considered for risk of bias and patterns of intervention effects. A randomized effect model was used to evaluate the dysbiosis of circulating microbiota and clinical outcomes. We conducted a meta-analysis considering the circulating bacteria in both healthy people and people with cardio-metabolic disorders, in reports published mainly from 2008 to 2022, according to the PRISMA guidelines. Results: We searched 627 studies and, after completing the risk of bias and selection, 31 studies comprising of 11,132 human samples were considered. This meta-analysis found that dysbiosis of phyla Proteobacteria, Firmicutes, and Bacteroidetes was associated with metabolic diseases. Conclusions: In most instances, metabolic diseases are linked to higher diversity and elevated bacterial DNA levels. Bacteroides abundance was higher in healthy people than with metabolic disorders. However, more rigorous studies are required to determine the role of bacterial dysbiosis in cardio-metabolic diseases. Understanding the relationship between dysbiosis and cardio-metabolic diseases, we can use the bacteria as therapeutics for the reversal of dysbiosis and targets for therapeutics use in cardio-metabolic diseases. In the future, circulating bacterial signatures can be used as biomarkers for the early detection of metabolic diseases.

## 1. Introduction

Despite the remarkable innovations in medical therapy, cardio-metabolic diseases (CMDs) remain one of the leading causes of morbidity and mortality globally, with maximum CMDs deaths reported from China during the last 5 years. By 2030, it is expected that 40.5% of the people living in the US will have some kind of cardiovascular disease (CVD) [1]. Non-infectious diseases cause millions of deaths every year, with 32% (WHO 2019) of deaths (only due to CVDs) occurring worldwide, costing an estimated $6.3 trillion and expected to double by 2030 [2]. CMDs, including cardiovascular diseases (CVDs), nonalcoholic fatty liver disease (NAFLD), type 2 diabetes (T2D), and chronic kidney disease (CKD), are a group of diseases with almost similar causes and symptoms [3]. Insufficient physical activity and excessive calorie consumption are hallmarks of the modern lifestyle, which has been linked to the development of CMDs [4]. Obesity and chronic low-grade inflammation brought on by a high calorie diet are precursors to metabolic syndrome, T2D, NAFLD, and CVDs [5,6,7,8]. A well-known factor in the progression of obesity to CMDs is the accumulation of ectopic fat, such as visceral adipose tissue with increased lipolysis, which causes an increased fatty acid influx in muscle cells. Ultimately, this disrupts mitochondrial insulin signaling, which leads to insulin resistance [9]. White adipose tissue in excess causes low-grade inflammation which leads to T2D, NAFLD, and CVDs [10]. Multiple factors contribute to CMDs progression. However, CMDs that share risk factors and characteristics may respond similarly to some treatments due to the similarities in their pathophysiological processes. However, unknown factors leading insulin resistance to T2D and CVDs remain to be clarified as traditional risk factors (e.g., dyslipidemia) have largely failed to account for the increased onset of CMDs and low response to treatments. The procedures for the excretion of uremic toxins and optimization of dialysis [11,12] are just two examples of many strategies that have been employed to lessen the inflammatory burden.

Omics technology revealed that chronic low-grade inflammation, as indicated by slight elevations in systemic inflammatory markers, e.g., C-reactive-Protein and lipopolysaccharides (LPS) [13], are widely thought to contribute to the emergence of cardio-metabolic dysfunction, insulin resistance, and diabetes [14,15]. So, it is crucial to know core blood microbial signatures in health and their dysbiosis (a state of deviation from a core functional and taxonomic composition) in disease conditions. It is well known that human microbiota (gut, oral, vaginal, and skin) is a contributing risk factor for cardio-metabolic diseases [16]. Several diseases, including obesity, diabetes, and cardiovascular disease, have been linked to human microbiota dysbiosis. Whole metagenome shotgun sequencing made it easy to detect the blood bacterial dysbiosis in health and disease but its use is still debated [17]. Cumulative evidence for bacterial signatures in metabolic tissues, including blood, recently opened new paths for exploring the potential impact of circulating microbiota on CMDs [18,19,20,21].

Microbial translocation into the bloodstream can occur through a variety of routes, including contaminated medical equipment, catheter use, and an impaired intestinal barrier, as well as scratches, insect bites, wounds, or mucosal surface exposures [19], and may cause chronic inflammation in patients with CMDs [22]. Detection of highly diverse microbial taxa in the systemic circulation, also known as the “circulating microbiome” [23], is a result of recent advances in sequencing technology that have allowed researchers to examine bacterial endotoxins (i.e., lipopolysaccharide) and other blood-borne microbial components [24,25]. Researchers also consider transient bacteremia as oral bacteria detected in the blood after teeth brushing or gingival manipulation [26,27]. There is mounting evidence of the persistent presence of microbial signatures among various patients’ blood without overt infections [28,29], even among the seemingly healthy individuals [30,31,32]. More recently, changes in the quantity and quality of circulating microbes and microbial products have been linked to the pathogenesis of chronic inflammation-related conditions, e.g., cardiovascular and metabolic disease [33,34]. These changes may play a role in the development of these diseases due to their immunostimulatory, atherogenic, and cardiotoxic properties [35,36].

Here, we presented the results of a meta-analysis and a systematic review of sequence-based studies on normal blood microbiota and its dysbiosis in people with cardiovascular disease and diabetes. Despite substantial variations in study design, we have been able to identify the circulatory microbial signatures that are consistent with and predictive of CMDs. This analysis utilizes the statistical power of the selected previous studies to define a reproducible bacterial dysbiosis during CMDs and demonstrate consistent computational tools across datasets, partially accounting for inter study differences. This meta-analysis will highlight some of the promising links between circulating microbiota and the risk of CMDs, and it will help us to better understand the role of the circulating microbiota as an influential factor for the progression of CMDs with clinical and research implications from this rapidly evolving field.

## 2. Materials and Methods

### 2.1. Search Strategy

In this study, we followed the PRISMA (Preferred Reporting Items for Systematic Reviews and Meta-analyses) guidelines [37]. A systematic search of English-language publications published from 2008 to 2022 was conducted via PubMed, Scopus, Medline, and Web of Science for studies published prior to 30 November 2022. Both medical subject heading (MeSH) terminology and free text words (Blood Microbiota OR circulating bacteria, OR transient bacteremia, OR cardiovascular disease, OR diabetes etc. search strategy is attached as Appendix A) were included. The target of literature search was the clinical case or cohort studies used blood or serum microbiome analysis via 16S ribosomal RNA sequencing or High throughput RNA sequencing, Illumina MiSeq, Shotgun Meta genomic sequencing of cell-free DNA and pyro-sequencing, etc.

### 2.2. Eligibility and Exclusion Criteria

Eligible studies were only the published studies which provide the 16S rRNA seq, Illumina MiSeq, Shotgun Meta genomic sequencing of cell-free DNA and pyro-sequencing, etc. Studies of cardio-metabolic patients using blood, blood plasma, or blood fractions as samples, which were observational in nature, with either cohort or case-control designs or retrospective data analysis, were considered for inclusion. Sepsis patients’ outcomes should have been studied with CVDs, CMDs and diabetes status. Animal studies were excluded. Studies using the prebiotics, probiotics, or any other treatment were excluded. Studies related to the assessment of skin or gut microbiota using blood samples were excluded. Furthermore, studies on cardio-metabolic diseases that lacked comparative data were excluded. Moreover, the studies focusing any other diseases, such as cancer, immune response, or infectious diseases, were excluded too.

### 2.3. Data Extraction and Quality Analysis

Two independent reviewers followed a pre-determined guide sheet to extract relevant data from studies that passed the evaluation’s eligibility and exclusion criteria (Figure 1 showing studies number and databank). We extracted the following information from each study (authors’ names, year of study, subject characteristics, sample size, origin of sample, country, microbiota sequencing methodology, and key circulating microbial findings) and used it to compile the data set. All the extracted data were cross checked by another independent reviewer and disagreements were discussed by all authors to achieve consensus. All discrepancies were dealt with according to the exclusion criteria mentioned above. The Newcastle-Ottawa Scale for Study Quality was used to evaluate the quality of the studies (≥7 score was considered for good quality methodology) [16]. The Cochrane Risk of Bias (RoB 2) [38] tool for randomized studies (Low, low risk of bias; Some, some arguments for risk of bias; High, high risk of bias) was also adopted.

### 2.4. Statistical Analysis

The meta-analysis was performed in R 4.1.3 (2022) with the appropriate base function unless otherwise stated along with 95% confidence intervals (CI). When possible, individual data points were displayed, and the mean and standard deviation have been shown elsewhere. The raw data were extracted from all relevant studies. Further, we divided the data on the basis of healthy and diseased subgroups (diabetic, cardiovascular, and miscellaneous). The analysis was conducted using the healthy individual as a normalized blood microbiota. While studies analyzing the dysbiosis of circulating microbiota in cardio-metabolic diseases were considered as a deviation of blood microbiota from healthy individuals. Unless otherwise specified, a *p* value of ≤0.05 was used to indicate statistical significance. Heterogeneity was analyzed by *I*^2^ statistics (>50% considered representative of statistical heterogeneity). A meta-regression and subgroup analysis was carried out to identify the possible sources of heterogeneity according to study location, methodology, disease status, and sample source for microbial abundance. The stability of the results was assessed by sensitivity analysis, by removing 1 study at a time from primary analysis.

## 3. Results

### 3.1. Study Selection and Exclusion

A search of databases of selected scientific forums found unique 627 articles following the screening of duplicate studies it reduced to 456. On the basis of title screening, 92 studies were excluded, and 289 papers were removed after abstract review. Further, among the remaining 75 studies, 36 were excluded following the full text review (Figure 1). The most important reasons for exclusion were wrong publication type (e.g., narrative reviews), gut, oral, or skin microbiota, infectious diseases, and samples other than blood. Then, 39 full-text articles were screened for eligibility, and eight studies (reviews narratives etc.) were excluded, leaving 31 human studies that met all inclusion criteria for the meta-analysis study. Inclusion and exclusion of studies was reviewed by the scientist working in the area of microbiome.

Among these studies seven out of 31 (22.5%) were conducted in 199 healthy individuals, diabetes studies (six out of 31 (19.35%)) involved 4838 diabetic and 154 healthy controls, and cardiovascular diseases (eight out of 31 (25.8%)) comprised of 4795 CVDs and 477 healthy controls. Meanwhile, 10 out of 31 (32.25%) studies considered miscellaneous metabolic diseases (metabolic disorders 517 and 142 healthy controls). The majority of the samples were extracted from whole blood. However, plasma, mononuclear cells, and blood fractions were also used to collect the bacterial DNA.

### 3.2. Quality Assessment of Included Studies

The selected 31 studies were analyzed for quality and risk of bias by using the guideline of Cochrane revised RoB 2 tool for randomized trials [38] and Newcastle-Ottawa Scale for non-randomized trials [39]. The majority of the clinical studies conducted on diseased and healthy controls were found to have an overall high risk of bias (Figure 2).

The domain 5 (selection of the reported outcome) of all studies was scored high risk as none of the selected studies reported a comprehensive, pre-specified plan for blood microbiome analysis. So, it is likely that results have been selected based on multiple eligible analyses of the data. Nonrandomized studies presented average scores for the selection criteria of the study participants, controlling for confounders (Table 1). No nonrandomized study with score above seven was considered for good methodological planning.

### 3.3. Circulating Bacterial Signatures in Healthy Individuals

The presence of live bacteria or other microorganisms in the blood of humans is an indication of sepsis. However, due to advances in the molecular detection of microbes, the term atopobiosis (presence of microorganisms within the blood of individuals without showing any disease) was proposed by Potgietor et al. [40]. Following the eligibility criteria, we identified seven studies which achieved the complete selection criteria. Among these, six were randomized studies having internal controls while one study was non-randomized and had no controls. The study participants ranged from two [41] to a maximum of 60 healthy individuals [42]. Out of seven studies, five studies used whole blood [31,41,43,44,45] as samples while two studies used fractions of blood [30,42] (plasma, RBCs, buffy coat) for microbiome detection. We focused on the studies having healthy individuals as main samples and internal controls. A majority of the studies analyzed the V3–V4 hyper-variable regions of the bacterial genome. Only two studies used cultural and sequence-based detection while other studies only focused on circulating bacterial DNA. Certain studies used the sequencing of bacterial rDNA or rRNA (RNA-seq) to evaluate the profiles and two studies used PCR and microscopy techniques (Table 2). Studies which used the cultural techniques to grow the bacteria from blood samples and sequencing are the most important as Damgaard et al. and Païssé et al. used fractions of blood to check the influence of transient bacteremia by analyzing the blood cell bacterial populations.

On the other hand, Figure 3 displays an average of the microbial abundance data of healthy individuals. Proteobacteria, with an average of 60% (ranging from 8–99%), are the most prevalent phyla among healthy people [47,61]. Actinobacteria are the second most prevalent phyla, with average of 12.6% (0.1–76%) [47,57], followed by Firmicutes with an average of 13.51% (0.13–44%) [47,61] and Bacteroidetes with an average of 7.14% (0.01–19%) [35,61]. These four phyla share more than 90% of blood circulating bacteria among healthy individuals. Among the others, Fusobacterium, Acidobacteria, Cyanobacteria, and Verrucomicrobia are the most prominent phyla. No abundance difference was found among the different bacterial phyla while using the amplicon-based or shotgun sequencing.

### 3.4. Circulating Bacterial Dysbiosis and Diabetes

Six of thirty-one studies focused on the T2D comprising of 4992 individuals. Out of six studies, four studies have a randomized controlled population [47,49,51,59] while one study has no controls [50]. Five studies used different variable regions and sequencing methods using the rRNA while Amar et al. used rDNA [50]. Two studies used the CARD-FISH visualization along with sequencing of the V4-V5 regions [48,51]. Six of the selected studies focused on T2D, and to our knowledge there is no study reporting the dysbiosis of circulating microbiota on type 1 diabetes. Studies included for diabetes had duration from a few days to nine months, with observational data from participants ranging from 75 to 3280. A nine-year study of over three thousand T2D patients found that a higher concentration of 16S rRNA was present in T2D patients, which could be the root cause of disease. In this meta-analysis, we found that a higher level of bacterial DNA was related with type 2 diabetes [29,46,50]. Furthermore, reversing the trend of increased Proteobacteria and decreased Actinobacteria in T2D patients can decrease the risk of type 2 diabetes [46]. The highly conserved regions of 16S rRNA subunits of the bacterial genome or other sequencing techniques can distinguish bacterial DNA fragments from human DNA [64].

Patients with T2D exhibited blood dysbiosis characterized by a decrease in orders (Rhodospirillales and Myxococcales) [18], genus (*Lactobacillus*, *Acinetobacter* and *Lactococcus*) [48], and loss of families (Bacillaceae and Bukholderiaceae) [51] compared to healthy individuals. Furthermore, dysbiosis of blood bacteria in the patients with type 2 diabetes showed higher abundance of *Alishewanella*, *Actinotalea*, *Pseudoclavibacter*, *Sediminibacterium* [47], *Tahibacter* [48], *Clostridium coccoides,* and the *Atopobium* [29] and *Ralstonia* spp. [46] which may increase one’s risk. Pre-diabetic and patients with morbid obesity tend to show lower abundance of *Akkermansia*, and *Faecalibacterium* (Table 2) compared to healthy individuals [49]. However, *Bacteroides* was found in abundance in people who were statistically less likely to develop T2D [47], suggesting that *Bacteroides* may act as a buffer against the development of T2D. Changes in T2D exhibited, with few exceptions, a similar pattern, with an average 17% relative increase in Proteobacteria and a nearly 11% decrease in Actinobacteria (Table 2).

### 3.5. Circulating Bacterial Dysbiosis and Cardiovascular Diseases

Traditionally, rheumatic heart disease and infectious carditis have been linked to blood microbial profiles. The present analysis shows that eight out of 31 studies were related to circulating bacteria altered in 4795 patients with all forms of heart diseases and 477 healthy controls. The duration of studies ranges from minimum 10 days [55] to nine and half years [54]. Most studies (six out of eight) were of randomized and comparative controls and two studies by Amar et al. [36,52] were interpretational studies. Minimum study population size was 31 and maximum was 3936 CVDs patients. All studies followed the qPCR and 16S rDNA or rRNA sequencing of different variable regions to evaluate the circulating microbiota. Two studies used the Ion Torrent PGM [33,35] method for sequencing while one of these used shot gun sequencing [35]. All participants had a previous report of cardiac events except the healthy controls. Population based findings and comparative studies based on 16S rRNA quantification identified that the patients with cardiac diseases had a higher concentration of 16S rRNA [52], increased abundance level of Proteobacteria [36,56], and lower abundance of Firmicutes [33] as compared to controls independently of other conventional risk factors (Figure 3). In a more than nine-year follow up study, a higher abundance of *Staphylococcus*, *Kocuria*, and *Enhydrobacter* and low levels of *Paracoccus* were associated with CVDs. Meanwhile, *Streptococcus*, *Paracoccus*, *Veillonella*, and *Bacteroides* were abundant in the healthy individuals [54]. In another study, cholesterol degrading genera (*Gordonia*, *Propionibacterium*, *Chryseobacterium*, and *Rhodococcus*) were depleted in the patients who developed myocardial infarction [52] while studies showed that individuals with higher abundance of *Bacteroides* are less prone to CVDs [54,55]. Blood dysbiosis in the hypertensive patients may be characterized by an increase in Proteobacteria but a decrease in Firmicutes and Bacteroidetes compared to healthy controls. Furthermore, Staphylococcus may act as a barrier and *Acinetobacter* or *Sphingomonas* may be risk factors for hypertension [53]. Overall, the results indicate that species related with cholesterol degradations and *Bacteroides* can be preventive tactics against CVDs.

### 3.6. Circulating Bacterial Dysbiosis and Miscellaneous Non-Infectious Diseases

We found 10 studies related to miscellaneous metabolic diseases which fulfilled the selection criteria. A total of 564 participants (517 diseased and 142 healthy controls) were recruited and population size was from seven to 108 participants. The sample size of these studies was small. Among the 10 selected studies, three studies used the sequencing bacterial culture techniques for the detection of circulating microbiota in cohorts [23,32,57]. The remaining seven studies used sequencing methods for the detection of different variable regions. Time duration for these studies ranged from three months to two years. In a pilot based study, increased bacteria 16S rDNA was related to liver fibrosis and proportions of Proteobacteria (specifically Sphingomonas, Variovorax, Bosea axa) in NAFLD [28]. Cardiac patients with sepsis following the cardiac surgery procedures had higher abundance of Proteobacteria vs. healthy individuals while Actinobacteria decreased in sepsis patients. Healthy individuals presented a greater predominance of Bifidobacteriales than the sepsis patients [57]. Likewise, the increased microbial biodiversity of 16S rDNA was observed in the patients with pancreatitis [58] and schizophrenia [59]. Patients with pancreatitis had higher abundance of Bacteroidetes and a decrease in Actinobacteria [58]. Genera, specifically *Bacteroides*, *Stenotrophomonas*, *Serratia*, *Rhizobium*, *Prevotella*, *Staphylococcus*, and *Paracoccus*, grew in patient blood regardless of illness severity, while *Acinetobacter*, *Lactococcus*, *Dietzia*, *Flavobacterium*, *Pseudomonas*, *Corynebacterium*, *Sphingobium*, and *Brevundimonas* were in low abundance [58]. Patients suffering from schizophrenia have higher microbial biodiversity in blood. In contrast, a recent pilot study showed that chronic kidney disease (CKD) has also been linked to the decreased alpha diversity of circulating bacteria, which is a hallmark of several non-communicable diseases. Enterobacteriaceae and Pseudomonadaceae are both significantly higher in CKD than HC (10% vs. 7%; and 23% vs18%, respectively [60]). In liver cirrhosis, positive cultivation of *Staphylococcus* and *Acinetobacter* was correlated with inflammatory cytokines [23]. While in some cases, Enterobacteriaceae were more prevalent, and a low abundance of *Akkermansia*, Rikenellaceae, and Erysipelotrichales was observed in LC patients [61]. In RA, blood dysbiosis increases *Halomonas*, *Anaerococcus*, *Shewanella*, members of Lachnospiraceae [62] *Pelagibacterium*, *Candidatus Saccharibacteria*, and Hyphomicrobiaceae, while a decrease in Bacteroidetes may play a role in rheumatoid arthritis pathophysiology [63].

## 4. Discussion

In this meta-analysis study, the presence of blood microbiota evaluated in healthy and metabolic diseases and presence of microbiota were screened based on bacterial genetic material (16S rDNA or rRNA variable regions) sequencing. According to our knowledge, this is the first meta-analysis study for circulating microbiota in healthy and CMDs conditions. However, reasons for CMDs are still unknown and the circulating microbiome link with CMDs is very limited. Blood contains bacteriostatic and bactericidal components that make it a hostile environment for microbes [65]. Previously, blood contamination during collection and experiments was considered accidental. Growing evidence suggests that bacteria or LPS binding protein (LBP) or LPS in the circulation cause systemic inflammation in cardio-metabolic diseases [17]. Several studies have added scientific reasoning and findings on the existence of a healthy human blood microbiome (Table 2) [66]. Different studies have found bacterial 16S rDNA or rRNA in blood of healthy individuals [31,44], healthy blood donors [30], and various patient populations without overt infections [36,53,54,56] by advanced and reliable sequencing techniques. Same techniques were used to detect archaea, fungi, and viruses from healthy people’s blood [44,67]. The presence of circulating bacteria reported in these studies is largely based on 16s rDNA or rRNA while few added viable bacterial cultures directly [32,44,57].

Our study found that the Proteobacteria is the most abundant phyla among the healthy subjects, followed by Firmicutes, Actinobacteria, and Bacteroidetes (Figure 4), and comprise more than 90% of the total blood microbiota, consistent with a variety of detection techniques ranging from cultural detection [23,32,44,57], qPCR [36,58], and shotgun sequencing [35], to pyro and Miseq sequencing of rDNA or rRNA [30,47]. Based on studies, minor variations can be considered due to age, location, climate, sex, and even occupation of individuals, which may affect the profile of the blood microbiome. The types of blood specimen (whole blood, plasma, buffy coat, or blood cells) may affect the phyla abundance among different studies (Figure 4). As the innate immune response of leukocytes and circulating DNase may remove bacterial cell-free DNA, so the blood bacteriome profile from blood cells may not reflect the true ecology of the blood bacteria [48].

A key finding of this study was comparative dysbiosis of circulating bacteria in the healthy and CMDs conditions (Figure 2). The bacteriome of plasma or serum might be represented by extracellular vesicles (EV) or bacterial cell-free DNA, which might not be eliminated by the circulating DNase. Firmicutes can outnumber the Proteobacteria and Actinobacteria if DNase is used. [68]. So, the studies based on plasma or serum fraction were inconclusive. In few studies, Firmicutes was the most abundant phylum [69], while others found a majority of Proteobacteria instead of Firmicutes [32,62,70]. The presence of bacteria in blood samples was selectively variable in each type of blood fractions, which is questionable, and sequencing techniques can also represent a variable factor. Therefore, we can assume that the whole blood samples and highly sensitive sequencing techniques can present better abundance for the blood bacteriome, containing all fractions of blood.

Our meta-analysis showed that the overall bacterial diversity was increased in patients with T2D and that people with T2D have higher levels of Proteobacteria (83%) as compared to healthy and overall CMDs (61%). Proteobacteria were consistently higher than healthy individuals in all studies with lower abundance of Firmicutes (Table 2 and Figure 4). While Firmicutes are more likely than healthy people to have bacteremia of unknown origin [71]. Circulating dysbiosis of several genera, specifically *Sediminibacterium*, *Tahibacter*, *Clostridium coccoide*, *Ralstonia* spp., *Akkermansia*, *Lactobacillus*, and *Faecalibacterium* (Table 2), was linked with T2D. Healthy individuals had higher abundance of Bacteroides (13%) compared to T2D (1%), which suggests that these genera can potentially be helpful in the reversal of diabetes or can act as a buffer. Fecal samples showed lower abundance of Firmicutes and reduced Firmicutes to Bacteroidetes ratio in studies related to diet composition and its effect on gut microbiota in T2D [72,73,74]. While studying the bacterial dysbiosis, we recommend recording the food habits of individuals under study as higher level of glucose and high fat diets can impact the dysbiosis. Circulation dysbiosis and dietary interventions may exacerbate chronic systemic inflammation, resulting in metabolic abnormalities such as diabetes. More research is needed to understand the circulating microbial dysbiosis of the phylogenetic clades and gene functions that we have linked to diabetes.

Historically, the cardiac complications associated with blood microbe profiles were those caused by infection only, e.g., rheumatic heart disease and infectious peri-, myo-, and endocarditis, which was primarily diagnosed using conventional serological and/or culture-based techniques. Conclusions drawn from our meta-analysis demonstrate that dysbiosis of the circulating microbiota is associated with CVDs. Long follow-up [36,54] and short term comparative studies [55] have shown that the Proteobacteria phylum average percentage was lower in CVDs (50%) in comparison with healthy (61%) counterparts, and Actinobateria (21%) were significantly higher than healthy controls (Figure 3). The blood of patients with cardiovascular disease was predominate with phylum Proteobacteria, with unique pro-inflammatory and pro-atherosclerotic properties that may contribute to the development and progression of cardiovascular disease [75]. Meanwhile, dysbiosis of cholesterol degrading bacterial Norcardiaceae and Aerococcaceae families and *Gordonia*, *Propionibacterium*, *Chryseobacterium*, and *Rhodococcus* genera [52] were lower in patients with CVDs; raising metabolic concerns of dysbiosis. Atherosclerotic plaques harbor bacterial DNA, primarily from Proteobacteria, and the same bacteria have been found in the guts of the same individuals [76], suggesting a breakdown in the intestinal epithelial barrier and correlating with studies based on blood microbial analysis.

Inflammation disrupts intestinal epithelial tight junctions and mucus-secreting goblet cells and mucosal lymph cells, releasing resident microbes into the bloodstream [67]. More studies are required to uncover this aspect of microbes’ translocation to the blood stream in CVDs. Included studies have not reported any bacterial product or inflammatory factor that can be a factor or outcome of dysbiosis. Two studies found that a higher abundance of Proteobacteria was observed in gut microbiota of coronary artery disease [77], hepatic steatosis [78], with lower abundance of Bacteroidetes. The decreased abundance of *Bacteroides* was a fascinating finding linking T2D and CVD. More intriguingly, research on the reversal of Proteobacteria dysbiosis and cholesterol degrading bacteria can be beneficial for people suffering from vascular diseases, such as ischemic heart disease and stroke. For CVDs dysbiosis, it is noteworthy to consider the occupations, living habits, and diet of the patients.

Studies focusing on other metabolic diseases are limited and most observed a small number of participants. Specific microbial signatures are associated with both hepatic steatosis and fibrosis [78,79]. We identified a study by Kajihara et al., 2019, which identified liver cirrhosis circulating microbiota in hepatitis, autoimmune, and other non-infectious factors. Phyla abundance of this study showed highly variable results compared to all other studies with high expression of Firmicutes of approximately 50% [61] which can be due to autoimmune and infectious reasons. This finding suggests that dysbiosis of infected individuals is different from that of the metabolic disorders. While other studies conducted on liver cirrhosis had higher abundance of circulating 16S rDNA and Proteobacteria (specifically *Sphingomonas*, *Bosea*, and Bradyrhizobiaceae taxa) were reported in liver patients [28]. This was further supported by studies that found the same abundance of liver cirrhosis in other blood compartments [23] and by a higher concentration of bacterial DNA via in situ hybridization. Meanwhile, Rikenellaceae and Erysipelotrichales were decreased in cases of liver cirrhosis [61]. Similarly, higher bacterial DNA levels were found in pancreatitis patients, with an increase in Bacteroidetes and Firmicutes. While the genera *Bacteroides*, *Stenotrophomonas*, and *Serretia* are more abundant, *Acinetobacter*, *Lactococcus*, *Flavibacterium*, and *Sphingobium* are less abundant in pancreatitis patients [58]. Chronic kidney disease (CKD), an established factor for CVDs, has been linked to changes in the blood bacterial signature, associated with decreased alpha bacterial diversity, and Enterobacteriaceae and Pseudomonadaceae were dominant in patients [60]. Bacterial dysbiosis is linked with an increase of indole- and p-cresyl-forming uremic enzymes, bacterial uremic toxins which are associated to worsening kidney function, endothelial dysfunction, cardiac fibrosis, macrophage activation, and insulin resistance [80]. Blood dysbiosis in RA is characterized by an increase in genera Halomonas, Anaerococcus, Shewanella, and members of Lachnospiraceae, while decrease in *Corynebacterium 1* and *Streptococcus* [62] and Bacteroidetes [63]. Gosiewski et al. (2017), Schierwagen et al. (2019), and Whittle et al. (2019) also tried to culture bacteria from blood samples and found some relevant results which suggest that more studies on developments for the culture of unculturable bacteria can be helpful for blood microbiota studies. No consistent dysbiosis of circulating microbiota was found in any disease condition, which may be due to the high risk of selection bias and heterogeneity of sample selection or study design. Considering all above studies, circulating bacterial dysbiosis, particularly higher bacterial diversity, is strongly linked with cardio-metabolic diseases.

Circulating bacterial dysbiosis relates to several CMDs. In this study, we observed that lower abundance of *Bacteroides* was reported in T2D patients, while dysbiosis of Proteobacteria and Actinobacteria were recorded in CVDs. A study by Kajihara et al. identified circulating microbiota in hepatitis, autoimmune, and other non-infectious diseases and suggested that dysbiosis of infected individuals is different from the metabolic disorders. However, detailed studies considering the advanced sequencing techniques and precise blood sampling can enhance our understandings of blood microbiota. Scientists must also consider the continuous uptake of microbiota from other body locations and transient bacteremia. Further, metabolomics studies are required alongside the genomic studies which can help to emphasize the bacterial metabolites responsible for CMDs. We found that Bacteroides can be used as buffer therapeutics in CMDs as their abundance was higher in healthy individuals. We also suggest that species related to cholesterol degradations and Bacteroides can be used as preventive tactics against CVDs and non-infectious diseases.

### 4.1. Study Strengths

To the best of our knowledge, this meta-analysis is the first to analyze the quantitative and systematic association between the circulating microbiota and cardio-metabolic diseases. It has several strengths such as the dysbiosis of different circulating phyla CMDs. Studies included in this meta-analysis are almost from across the world. Moreover, we conducted subgroup analyses and meta-regression to find the dysbiosis in diabetes, cardiovascular, and miscellaneous metabolic disorders. Our analysis showed that the presence of genomic bacterial DNA can be the key factor for chronic inflammation which is a key factor for CMDs.

### 4.2. Limitations

The present meta-analysis has some limitations. Studies included in our meta-analysis are limited as this concept is very new and most of the studies do not have negative controls for laboratory solutions or equipment. This study only analyzed the dysbiosis at phyla level and lacks detailed consideration of the dysbiosis at species levels. As most of the studies are based on genomic DNA and lack bacterial cultures from healthy or diseased individuals, which makes it difficult to decide whether the circulating bacteria are from the transient bacteremia, bacterial clearance by immune cells, or regular uptake of bacteria from other body parts such as gut or oral microbiota. So, it is difficult to make conclusive statement regarding which phyla play key roles in CMDs on the basis of available data.

## 5. Conclusions

Blood microbiota profiles might be used as a potential noninvasive biomarker for human critical cardiovascular and other diseases. However, it is not yet known whether alteration in the blood microbiome is just a bystander of dysbiosis or a true player in the pathophysiology of diseases. Previously, we reviewed that dysbiosis of human blood microbiota could be an outcome of cutaneous or mucosal barrier disturbance, transitional bacteremia following infectious diseases, as well as failure of bacterial clearance by the liver. Gut, oral, and lung microbiota may be the primary sources of the blood bacteriome [19]. Still, there are critics regarding the presence of blood microbiota. Due to increasing studies based on blood microbiota signatures in several comparative and only diseased individuals, we consider the presence of blood circulating bacteria. However, more research is needed to make it clear whether it is blood microbiota or transient bacteremia. In our meta-analysis, we considered it very important to consider negative and healthy controls while studying bacterial dysbiosis. The differential expression of tight junction proteins, tracking of Mucus-secreting goblet cells, and mucosal lymph cells translocation into blood stream can predict better understanding for the presence of blood microbiota. Experiments that require minimal handling and avoid skin, reagent, and equipment cross-contamination should be designed. Improved sample collection, processing, and data production can improve contamination free results. Further, we suggest to include the concepts of immuno-metabolism, a newly emerging research area, as dysbiosis is related with expression levels of different cytokines. The emerging area of tissue microbiome research is still in its infancy but promises novel prognostic, diagnostic, and therapeutic strategies with paradigm-shifting potential for health care and disease prevention and treatment.

## Figures and Tables

**Figure 1 microorganisms-11-00777-f001:**
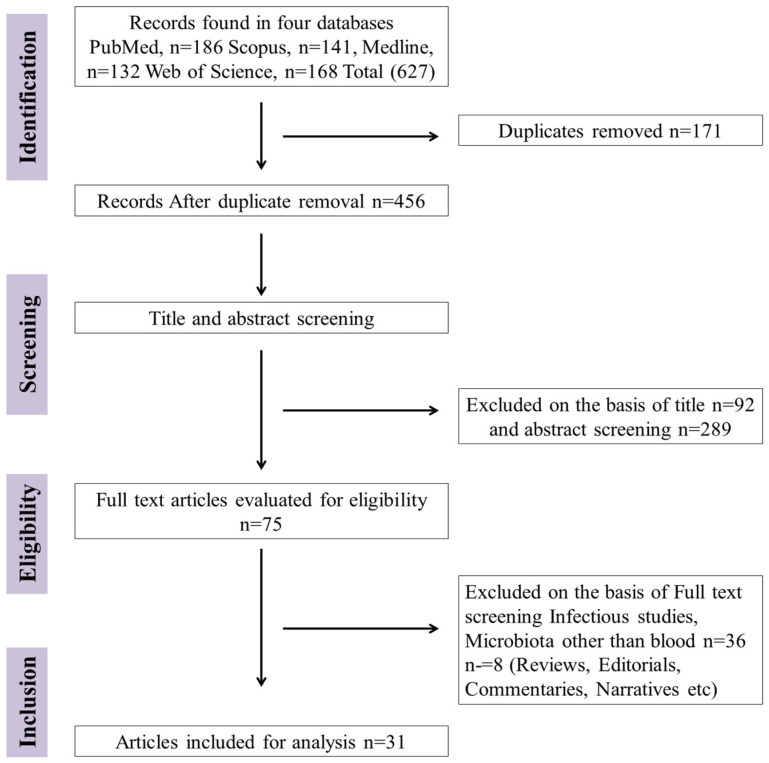
Flow diagram of studies searched in databases, screening exclusion and inclusion of studies (Preferred Reporting Items for Systematic Reviews and Meta-Analyses; PRISMA).

**Figure 2 microorganisms-11-00777-f002:**
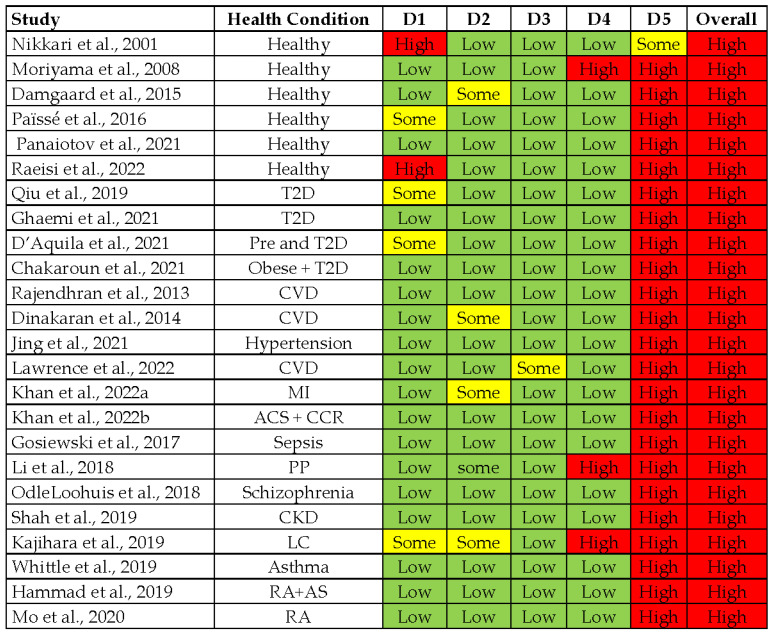
Assessment of study quality by using the Cochrane Checklist for Randomized studies. (Green is for Low, Yellow is for some and Red is for High).

**Figure 3 microorganisms-11-00777-f003:**
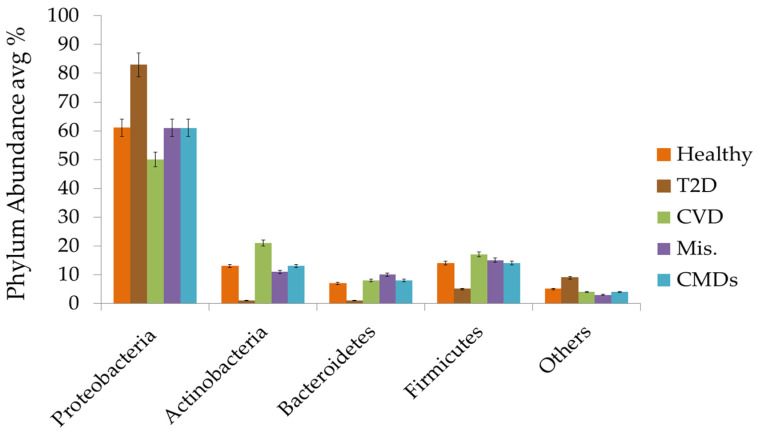
Average abundance of different phylum in healthy, T2D, CVDs, Mis., and overall in CMDs.

**Figure 4 microorganisms-11-00777-f004:**
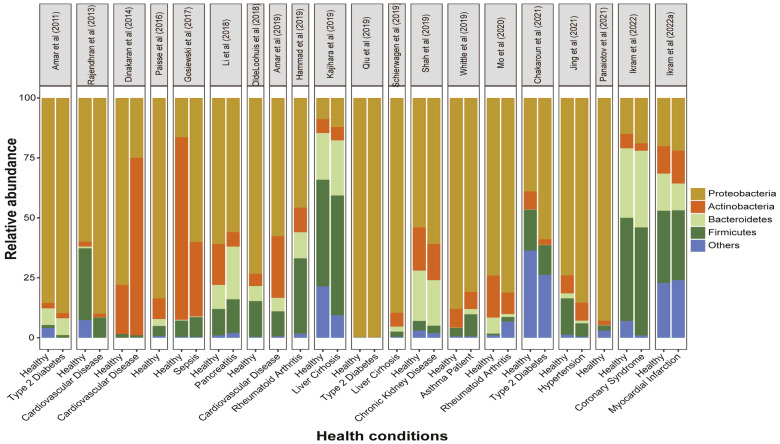
Phylum distribution in blood bacterial signatures from healthy and cardio-metabolic and cardiovascular disease cohorts. Each color represents a different phylum, and bar length equals the percentage of those phyla that were reported. Top pane is representing authors’ name and publication year. All studies comparing different diseases and only healthy individuals were considered.

**Table 1 microorganisms-11-00777-t001:** Risk of Bias for non-randomized studies according to Newcastle-Ottawa Scale for non-randomized trials.

Study	Selection (4*)	Comparability (2*)	Outcome (3*)	NOS Score
McLaughlin et al., 2002	* *		*	3/9
Amar et al., 2011	* * *	*	*	5/9
Massier et al., 2020	* *	*	*	4/9
Amar et al., 2013	* * *		* *	5/9
Amar et al., 2019	* * *		* *	5/9
Lelouvier et al., 2016	* *		* *	4/9
Schierwagen et al., 2019	* * *		* *	5/9

**Table 2 microorganisms-11-00777-t002:** Summary of different human blood microbiota studies conducted on healthy or diseased individuals.

Authors	Study Population	Sample	Study Design	Country	Methodology	Changes in Circulating Microbiota	Ref.
Healthy individuals							
Nikkari et al., 2001	4 HI	Whole Blood	Cohort Study	USA	qPCR and rRNA targeting the conserved region of 16S rDNA by fluorescent probe	Bacteria from 5 divisions and 7 distinct phylogenetic groups detected in the blood.	[31]
McLaughlin et al., 2002	25 HI	Whole blood	Cohort Study	Canada	Characterization by 16S rRNA and gyrB genes Dark-field microscopy and FISH.	Pleomorphic antibiotic susceptible bacteria existing in healthy blood with limited growth a (possibly Pseudomonas).	[45]
Moriyama et al., 2008	2 HI	Whole Blood	Preliminary Study	Japan	16S rRNA PCR and Sanger sequencing.	*Aquabacterium*, *Budvicia*, *Stenotrophomonas*, *Serratia*, *Bacillus* and *Flavobacteria* identified only in clones.	[41]
Damgaard et al., 2015	60 HI	Blood plasma and RBCs	Cross Sectional Study	Denmark	Blood samples incubated on TSA or blue lactose plates, 16S rRNA gene sequencing and colony PCR	Bacterial growth observed in 35% of RBC fractions and 53% of plasma fractions. Staphylococci, *Propionibacterium*, *Micrococcus* and *Bacillus* most frequently found.	[42]
Païssé et al., 2016	30 HI	Buffy Coat, Plasma and RBCs	Cohort Study	France	16S rRNA gene qPCR and 16S targeted metagenomic sequencing (Illumina MiSeq)	All blood fractions were positive for bacterial DNA. Most prevalent Phyla were Proteobacteria, Actinobacteria, Firmicutes and Bacteroidetes	[30]
Panaiotov et al., 2021	28 HI	Whole blood,	Cohort Study	Bulgaria	16S rRNA genes and ITS2 targeted sequencing on Illumina MiSeq and TEM.	Cultural and molecular characterization of healthy blood microbiota (Proteobacteria and Frimicutes were prominent)	[44]
Raeisi et al., 2022	50 HI	Whole blood	Cohort Study	Iran	16S rRNA gene PCR pyro-sequencing	Cultures positive (12%) PCR positive (12%) Staphylococcus, Bacilli were main findings Direct blood PCR-sequencing: *Bulkholderia*	[43]
Diabetic vs. healthy Individuals							
Amar et al., 2011	3280T2D	Leukocytes	Longitudinal cohort study	France	16S rDNA quantitative PCR and Pooled pyrosequencing (V1-V2)	High 16S rDNA levels caused diabetes despite risk factors. Proteobacteria phylum represented the highest relative abundance (~90%). High prevalence of *Ralstonia* spp. in diabetes patients.	[46]
Qiu et al., 2019	150 (50 T2D 100 HI)	Blood plasma	Nested case-control study	China	16S rRNA amplicon sequencing by Illumina MiSeq (V5-V6)	*Aquabacterium*, *Pseudonocardia*, and *Xanthomonas* genera and *Bacteroides* spp. showed an inverse while Alishewanella, *Actinotalea*, *Pseudoclavibacter*, *Sediminibacterium* spp. showed a positive correlation with diabetes.	[47]
Massier et al., 2020	75, (42 obese and 33 T2D)	Blood	Comparative Cohort study	Germany	16S rRNA gene sequencing (V4-V5), CARD-FISH	Genus *Lactobacillus*, *Acinetobacter* and *Lactococcus* decreased in T2D while *Tahibacter* increased in T2D	[48]
Ghaemi et al., 2021	90(30 T2D, 30 Pre-D and 30 HI)	Buffy Coat	Cohort Study	Iran	Real-time PCR using genus-specific 16srRNA primers	*Akkermansia*, and *Faecalibacterium* were higher in HI compared to pre-diabetic and T2D	[49]
D’Aquila et al., 2021	1285 MARK-AGE Study	Whole blood	Cross-sectional Study	Selected European Countries	Quantification of 16S rRNA by Real-time qPCR (V3-V4)	High level of Bacterial DNA were associated with higher level of Insulin and glucose	[50]
Chakaroun et al., 2021	112(64 BO 24 T2D and 24 HI)	Whole Blood	Cohort Study	Germany	16s rRNA sequencing (V4-V5) Bacteria were visualized by CARD-FISH	Loss of Bacillaceae and Bukholderiaceae in T2D and *Anoxybacillus*, *Duganella*, *Acidibacter*, *Chryseomicrobium, Sphingomonas* were decreased	[51]
Cardiovascular vs. Healthy Individuals							
Amar et al., 2013	3936 CVD	Leukocytes	Longitudinal Study	France	Eubacteria and Proteobacteria 16S rDNA by qPCR	There was a positive correlation of Proteobacteria, and vice versa of Eubacteria, with cardiovascular events.	[36]
Rajendhran et al., 2013	41 (31 CVD 10 HI)	Whole blood	Comparative Case Study	India	Amplicon sequencing of 16S rDNA V3 region (Ion Torrent PGM)	Increase of Proteobacteria (Pseudomonadaceae and Gammaproteobacteria), decrease in Firmicutes (Staphylococcaceae) and Bacillales, in CVDs	[33]
Dinakaran et al., 2014	120(80 CVD 40 HI)	Blood plasma	Comparative quantitative study	India	16S rDNA and β-globin gene concentrations by qRT-PCR. Shotgun and amplicon sequencing V3 region (Ion Torrent PGM)	The 16S rRNA/β-globin gene ratio was higher in CVDs patients than in controls Actinobacteria and Bacteriophages were dominant in CVDs patients whereas Proteobacteria and eukaryotic viruses were dominant in controls.	[35]
Amar et al., 2019	202 (99 CVD with MI 103 HRCVD)	Whole Blood	Case Control Study	France	16S rRNA sequencing for V3–V4 regions	Norcardiaceae and Aerococcaceae families and *Propionibacterium*, *Gordonia*, *Chryseobacterium*, and *Rhodococcus* genera (cholesterol-degrading bacteria) were lower in patients with (vs. without) myocardial infarction. An increase in 16S rRNA gene concentration in CVDs.	[52]
Jing et al., 2021	300 (150 Hypertension and 150 HI)	Blood Plasma	Case Control Study	China	16S rRNA gene (V6-V7) Miseq Illumina sequencing	*Staphylococcus* might be a protective factor for hypertension while either *Acinetobacter* and *Sphingomonas* might be are risk factor for hypertension	[53]
Lawrence et al., 2022	405 (CVD227, HI 178)	Whole Blood	Case Cohort Study	Norway	16S rRNA gene sequencing (V3-V5)	Higher abundance of *Staphylococcus*, *Kocuria*, *Enhydrobacter* and low levels of *Paracoccus* were associated with CVDs. While *Streptococcus*, *Paracoccus*, *Veillonella*, and *Bacteroides* were in abundance in HI.	[54]
Khan et al., 2022a	58 (MI 29 and 29 HI)	Whole Blood	Comparative Case control study	China	16S rRNA genes (V3-V4) Miseq Illumina sequencing	Decreased alpha diversity in MI patients. Abundance of Bifidobacterium and Bacteroides was increased in MI patients and HI respectively.	[55]
Khan et al., 2022b	210 (ACS, CCR and HI 70 each)	Whole Blood	Comparative Case control study	China	16S rRNA genes (V3-V4) Illumina sequencing	Proteobacteria and Desulfabacterota abundance was higher in ACS and Actinobacteria in HI	[56]
Miscellaneous diseases and Healthy							
Lelouvier et al., 2016	108 (NAFLD 22 with 86 without fibrosis)	Buffy Coat	cross-sectional study	France	16S rDNA PCR and (V-3V4) Miseq	Increased 16S rDNA and proportions of Proteobacteria (specifically Sphingomonas, Variovorax, Bosea taxa) in NAFLD	[28]
Gosiewski et al., 2017	85 (62 sepsis 23 HI)	Whole Blood	Comparative Cohort Study	Poland	16S rRNA gene targeted metagenomic Miseq Illumina (V3-V4) and Cultures	Healthy samples presented higher diversity than sepsis patients. Proteobacteria were lower in healthy individuals, while Actinobacteria decreased in sepsis patients. HI had predominance of anaerobic bacteria of order Bifidobacteriales	[57]
Li et al., 2018	62 (50 PP and 12 HI)	Whole blood and Neutrophils	Cohort Study	China	16S rDNA, qPCR and targeted metagenomic sequencing V3 region (Ion Torrent PGM)	16S rDNA gene copies were higher in patients. Bacteroidetes were high and Actinobacteria lower in patients.	[58]
OdleLoohuis et al., 2018	192 (48 SCZ, ALS 47, BPD 48, 49 HI)	Whole Blood	Cohort Study	USA	High Quality unmapped RNA sequencing	Proteobacteria, Firmicutes and Cyanobacteria were most prevalent phyla while Schizophrenia patients have diverse microbes. Bacteria and CD8+ memory T cells are inversely related.	[59]
Shah et al., 2019	40(20CKD and 20 HI)	Buffy Coat	Cross sectional Pilot Study	France	16S rRNA (V3–V4) regions gene sequencing	Enterobacteriaceae and Pseudomonadaceae significantly higher in CKD than HC (10% versus 7%; and 23% versus 18%, respectively.	[60]
Schierwagen et al., 2019	7 LC	Venous Blood	Cohort Study	Germany	16s rRNA Gene sequencing, Anaerobic Cultivation	Positive cultivation of *Staphylococcus* and *Acinetobacter*, Inflammatory cytokine positively correlated with abundance of blood microbiome	[23]
Kajihara et al., 2019	80 (66 LC and 14 HI)	Peripheral Blood	Cohort Study	Japan	16s rRNA Gene sequencing, (V3-V4)	Enterobacteriaceae was higher in LC. On the contrary, the abundance of Akkermansia, Rikenellaceae and Erysipelotrichales were lower in LC	[61]
Whittle et al., 2019	10 (5 Asthma and 5 H women)	Plasma fractions	Cohort Study	United Kingdom	Cultures an16S rRNA gene sequencing	Most abundant phyla were Proteobacteria, Actinobacteria, Firmicutes and Bacteroidetes. Achromobacter and Pseudomonas were decreased in Asthma patients	[32]
Hammad et al., 2019	32 (20RA, 4AS, 4 PA, 4HI)	Serum	Cohort Study	United Kingdom	16S rDNA sequencing V4 region	Blood dysbiosis in RA characterized by an increase in genera *Halomonas*, *Anaerococcus*, *Shewanella*, and members of Lachnospiraceae, while decrease in *Corynebacterium 1* and *Streptococcus*	[62]
Mo et al., 2020	43 (28 RA and 15 HI)	Mononuclear cells	Cohort Study	China	16S rDNA sequencing	Bacteroidetes had less abundance in RA while *Candidatus Saccharibacteria* was increased.	[63]

Healthy Individuals (HI), Diabetic Patients (DP), Non Diabetic Patients (NDP), Type 2 Diabetes (T2D), Baseline obesity (BO), Cardiovascular diseases (CVDs), Myocardial Infarction (MI), High risk Cardiovascular diseases (HRCVD), Coronary Syndrome (CS), Chronic Kidney Disease (CKD), Liver cirrhosis (LC), Coronary Heart disease (CHD), Rheumatoid arthritis (RA), Pancreatitis’ patients (PP),Schizophrenia (SCZ), Amyotrophic lateral sclerosis (ALS), Bipolar disorder (BPD), Ankylosing spondylitis (AS), Psoriatic arthritis (PA), Fluorescent insitu hybridization(FISH).

## Data Availability

Not applicable.

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
