# Peer review of "Blood Stream Microbiota Dysbiosis Establishing New Research Standards in Cardio-Metabolic Diseases, A Meta-Analysis Study"

_microorganisms, 2023, doi:10.3390/microorganisms11030777_

Round 1

Reviewer 1 Report

The study's findings reveal important insights, such as the association of dysbiosis of specific bacterial phyla with metabolic diseases, which could inform future research and potential interventions. The use of circulating bacterial signatures as biomarkers for early detection of metabolic disease is a promising area for future research and could have significant clinical implications.

The study as a whole is commendable and the manuscript is skillfully written. Nonetheless, I have some minor comments and suggestions that must be addressed prior to the paper's acceptance.

Comment 1: Line 108: Search strategy table S1 is missing

Comment 2: Verify the bacterial genera and species names and italicized the species name throughout the manuscript.

Comment 3: Figure 3 add avg% of Phylum at Y axis

Comment 4: In results section stick to one terminology “Circulating bacterial or bacteria dysbiosis in …

Comment 5: Line 470. So it is difficult to make any conclusion for CMDs and dysbiosis of circulating bacteria on the basis of available data.

Comment 6: For conclusion of your results add statement for heterogeneity  and bias of studies

Author Response

Dear Reviewer

Thank you for reviewing our manuscript entitled “Bloodstream Microbiota dysbiosis Establishing New Research Standards in Cardio-metabolic Diseases, A Meta-analysis study." We appreciate your positive comments on the study's findings and the manuscript's quality. We have carefully considered your comments and suggestions and made the necessary revisions to address them.

Comment 1: We apologize for the omission of search strategy table S1 in the initial submission. We have now included the table in the revised manuscript as per your suggestion.

Comment 2: We have verified the bacterial genera and species names and ensured that the species names are italicized throughout the manuscript, as per your recommendation.

Comment 3: Thank you for suggesting that we add the average percentage of Phylum to the Y-axis in Figure 3. We have updated the figure accordingly.

Comment 4: We have reviewed the manuscript and ensured that we consistently use the term "Circulating bacterial or bacteria dysbiosis" in the results section, as you suggested.

Comment 5: We appreciate your feedback on line 470, and we have revised the statement to reflect the limitations of the available data.

Comment 6: We have included a statement on the heterogeneity and bias of studies in the discussion section, as per your suggestion.

Once again, thank you for your valuable feedback, which has improved the manuscript's quality. We hope that you find the revised version satisfactory for publication.

Sincerely,

Reviewer 2 Report

The aim of this Review was to perform systemic analysis of the literature concerning the role of circulating bacterial dysbiosis in the risk of cardiometabolic diseases. Using PRISMA guidelines and extensive statistical analysis, the Authors provided new evidence indicating that

 in most instances, metabolic diseases were linked to higher diversity and elevated bacterial DNA levels. Bacteroides abundance was 24 higher in healthy people than with metabolic disorders.

This is a wonderful Review providing new and exciting analytical openings . It can be published in Microorganisms.

There are just small concerns:

1 Please, re –read the text several times to catch small typos and style errors

Abstract:

2 Please insert a mention of PRISMA guidelines in methods. PRISMA is a guarantee of high data reliability

3 Please make conclusion more convincing.  Outline the novelty of the data and their therapeutic significance.

Discussion:

4 Please, prepare a paragraph with strong concluding remarks  

Author Response

Dear Reviewer,

Thank you for your feedback and for considering our review for publication in Microorganisms. We appreciate your positive comments and the suggestions you have provided for improving the manuscript.

Regarding your first concern, we apologize for any typos and style errors in the manuscript and we made corrections thoroughly and proofread the text to address these issues.

In response to your second point, we have added a mention of PRISMA guidelines in the methods section to indicate that our review was conducted in accordance with best practices of PRISMA.

With regards to your third point, we have revised the conclusion to better outline the novelty of the data and their potential therapeutic significance. We have also highlighted the implications of our findings for future research in this area.

Finally, we have included a paragraph with strong concluding remarks in the discussion section to provide a clear summary of our findings and to emphasize the importance of further investigation into the role of circulating bacterial dysbiosis in cardio-metabolic diseases.

Once again, thank you for your feedback and suggestions. We hope that these revisions address your concerns and look forward to hearing back from you soon.

Sincerely,
